# Eco-Geography and Phenology Are the Major Drivers of Reproductive Isolation in the Royal Irises, a Species Complex in the Course of Speciation

**DOI:** 10.3390/plants11233306

**Published:** 2022-11-29

**Authors:** Inna Osmolovsky, Mariana Shifrin, Inbal Gamliel, Jonathan Belmaker, Yuval Sapir

**Affiliations:** 1The Botanical Garden, School of Plant Science and Food Security, The George S. Wise Faculty of Life Sciences, Tel Aviv University, Tel Aviv 69978, Israel; 2School of Zoology, The George S. Wise Faculty of Life Sciences, Tel Aviv University, Tel Aviv 69978, Israel

**Keywords:** species complex, ecological speciation, speciation continuum, reproductive isolation

## Abstract

The continuous nature of speciation implies that different species are found at different stages of divergence, from no- to complete reproductive isolation. This process and its underlying mechanisms are best viewed in incipient species. Moreover, the species complex can offer unique insight into how reproductive isolation (RI) has evolved. The royal irises (*Iris* section *Oncocyclus*) are a young group of species in the course of speciation, providing an ideal system for speciation study. We quantified pre- and post-zygotic reproductive barriers between the eight Israeli species of this complex and estimated the total RI among them. We tested for both pre-pollination and post-pollination reproductive barriers. Pre-pollination barriers, i.e., eco-geographic divergence and phenological differentiation were the major contributors to RI among the *Iris* species. On the other hand, post-pollination barriers, namely pollen–stigma interactions, fruit set, and seed viability had negligible contributions to total RI. The strength of RI was not uniform across the species complex, suggesting that species may have diverged at different rates. Overall, this study in a young, recently diverged group of species provides insight into the first steps of speciation, suggesting a crucial role of the pre-zygotic barriers.

## 1. Introduction

Speciation, the process by which species diverge from each other and maintain their boundaries, has been the focus of evolutionary research since Darwin’s “On the origin of species” [1,2,3,4,5]. Under the biological species concept, reproductive isolation (RI) between groups, or populations determines these boundaries [2,6]. Consequently, speciation is the process of accumulating reproductive barriers between populations [2,5], which are the primary units of speciation [7,8,9]. This is a continuous process, and species could be found at different stages of divergence, from full gene flow to complete isolation, or at equilibrium [8]. Incipient species are defined as a group of diverging species that maintain the substantial potential for gene flow due to incomplete RI [10,11,12]. The extent of gene flow between diverging taxa can be asymmetric [13,14,15]. Thus, asymmetric, continuous, and incomplete RI attests to the complex nature of speciation [4,16,17]. While well-defined species’ boundaries are at the final steps of the speciation continuum, studying the process in incipient species provides a better understanding of the evolution of species divergence.

Ample evidence across plant taxa suggests that most plant species are not fully reproductively isolated from one another [18], but that strong RI can result from either strong pre- or post-zygotic reproductive barriers. Pre-pollination barriers are considered to be stronger than post-pollination ones [3], mainly because of their early occurrence [5]. Several reproductive barriers acting together usually create additive effects contributing to the total RI [5,19,20,21].

Quantifying the extent of RI in a species complex can highlight a specific reproductive barrier that dominates the RI across the complex, or if different reproductive barriers co-act in the process of divergence. In a species complex, different species can be at different stages of divergence from one another [22]. Studying a complex of phylogenetically related species, provides insight into the interplay between the accumulation of reproductive barriers and genetic differentiation, by correlating RI with genetic divergence [23,24,25]. For example, [23] found a strong correlation between genetic divergence and RI in three genera, while two other genera exhibited only a weak correlation. In Mediterranean orchids, [9] found a correlation between evolutionary rates and post-zygotic barriers in food-deceptive species, while sexually deceptive species were isolated by pre-mating barriers, and the correlation with evolutionary rates was weaker. In a complex of diploid *Fragaria* species, [24] found that reproductive isolation is mainly governed by very late stages of post-zygotic barriers, and is not associated with genetic distance. In the species complex of Jewelflower, Christie and Strauss [25] found that the strength of intrinsic postzygotic reproductive barriers was positively correlated with the age of divergence between pairs of species. These studies suggest that genetic distance will be associated with reproductive isolation in a species complex.

We studied the extent of reproductive isolation within a species complex of the genus *Iris*. The royal irises (section *Oncocyclus* of the genus *Iris)* are a monophyletic young group, comprised of approximately 33 species distributed throughout the Middle East, with eight of the species endemic or sub-endemic to Israel [26,27,28,29]; (Figure 1). These species are *Iris atrofusca* Baker, *I. atropurpurea* Dinsm., *I. bismarckiana* Regel, *I. haynei* Baker, *I. hermona* Dinsm., *I. lortetii* Barbey, *I. mariae* Barbey, and *I. petrana* Dinsm. (*Iris* aff. *petrana* Dinsm.; [30]). These eight species are phenotypically and geographically distinct from each other. The eight Israeli species are phylogenetically clustered in one clade with species growing in the southern Levant (Israel, Jordan, Lebanon, and Syria), although without a clear clustering of specific geographic locations [29]. Morphological and genetic research revealed a continuous change along the climatic gradient, without strict borders among species [31,32]. A qualitative study in the 1970s showed that the species in this section are not reproductively isolated from each other and can produce viable and fertile hybrids [33], but no hybrids were recognized in the wild (Y. Sapir, personal observations). Partial post-zygotic RI among populations was shown within *Iris atropurpurea*, one of the species in the group [34]. Recent studies [35,36] have found no post-zygotic reproductive isolation, with eco-geography as the main barrier enforcing isolation between four species. Altogether, evidence supports the hypothesis that the royal irises are in the course of speciation [37]. Therefore, these plants can serve as a model system for studying the evolution of reproductive barriers, their magnitude, and their relative contribution to RI.

Here we report a study aimed to understand the speciation process, and its strength, in the royal irises. Our aim was to identify the reproductive barriers contributing the most to the speciation process in the royal irises, compare pairwise RI among species, and observe whether the speciation process was uniform within the group. Specifically, we hypothesized that (1) RI among the *Oncocyclus* irises is governed mainly by eco-geographic isolation; (2) pairwise genetic distance will correlate with RI; and (3) species pairs in the group will exhibit the same reproductive barrier and similar pairwise RI. To test these hypotheses, we quantified the pairwise reproductive barriers, both pre- and post-zygotic, between eight *Oncocyclus* species growing in Israel and Palestine. Pre-zygotic barriers include eco-geographical niche overlap, flowering phenology, pollen–stigma interactions, and post-zygotic fruit- and seed-set. We quantified total RI following Ramsey et al. [14] and Sobel and Chen [38], accounting for the sequence of potential barriers and their relative and absolute contribution to total RI (see below). Finally, we tested whether RI is correlated with genetic distance among species.

## 2. Results

### 2.1. Eco-Geography

The distribution of the potential spatial niches for all species is presented in Figure 2. The GLM model of species distribution for each species had high model accuracy scores, with AUC values ranging between 0.91 and 0.99 (median = 0.96) and TSS values ranging between 0.72 and 0.96 (median = 0.89; see Appendix A). The Niche overlap between species, estimated by the D index was 0.01–1 with a median of 0.15 (Table 1). The highest overlap values were among the northern species (*I. hermona*, *I. bismarckiana*, *I. lortetii*, and *I. haynei*). In addition, a high eco-geographic overlap was observed between *I. petrana* and the other two southern species: *I. mariae* and *I. atrofusca*. 

### 2.2. Phenology

Common garden: Flowering time overlap among the species in the net house varied, ranging from D = 0 between *I. hermona* and *I. lortetii* to D = 0.97 between *I. hermona* and *I. petrana*, with a median of 0.39. Most of the species growing under common garden conditions in TAUBG overlapped in their flowering time, except *I. atropurpurea* and *I. lortetii*, which flowered significantly earlier or later, respectively, compared to all the other species (time-to-event analysis: Z = 5.3, *p* < 0.001 and Z = −5.02, *p* < 0.001, respectively; Figure 3A). These differences were confirmed after controlling for the spatial location of the plants within the net house, which significantly affected flowering time (Z = 3.572, *p* < 0.001).Wild populations: D values of flowering time overlap among the species in the wild populations ranged between 0.067 (*I. atropurpurea* and *I. lortetii*) and 0.85 (*I. bismarckiana* and *I. lortetii*) with a median of 0.46. Comparison between the species observed in the wild populations revealed that *I. atropurpurea*, *I. lortetii*, and *I. petrana* significantly differed in flowering time from the other species (contrast analysis: Z = 11.351, *p* < 0.001; Z = −12.826, *p* < 0.001, and Z = −3.068, *p* = 0.002, respectively; Figure 3B). These differences were confirmed after controlling for the effect of the year of observations, which showed a significant effect (Z = 3.338, *p* < 0.001).

### 2.3. Pollen-Pistil Interactions

A total of 655 stigmas were treated over 2 months of this experiment. Acceptor species and the date of hand pollination significantly affected pollen germination (χ^2^_651, 653_ = 159.9, *p* < 0.001 and χ^2^_647, 648_ = 6.9, *p* = 0.008, respectively). Treatment, i.e., cross within or between species, did not significantly affect the fraction of pollen germination (χ^2^_648, 651_ = 5.8, *p* = 0.121; Figure 4). The covariance analysis revealed significant effects of the date (F_1, 648_ = 4.89, *p* = 0.027, germination data were arcsine transformed to improve normality) and acceptor species (F_2, 648_ = 3.6, *p* = 0.03), but pollen origin (within/between species) did not significantly affect the fraction of germination (F_1, 648_ = 1.35, *p* = 0.25). 

### 2.4. Post-Zygotic Reproductive Barriers—Cross Experiment

A total of 323 crosses were performed on the species. Of these, 47 (both within and between species) did not produce viable fruit. The covariance analysis revealed a significant effect of the acceptor species on both the fruit set and the proportion of viable seeds (F_5, 353_ = 4.97, *p* < 0.01 and F_5, 299_ = 4.82, *p* < 0.01, respectively, data for the seed set were arcsine-transformed to improve normality). *I atropurpurea* had both the highest fruit set and proportion of viable seeds, followed by *I. mariae* and then by *I. petrana* (Figure 5). Treatment, namely crosses within and between species, had no significant effect on both fruit set and viable seed proportions (F_1, 353_ = 1.93, *p* = 0.17 and F_1, 299_ = 0.53, *p* = 0.47, respectively).

### 2.5. Reproductive Isolation

Pre-zygotic pre-pollination barriers, namely eco-geographic barriers, had the highest contribution to reproductive isolation between all species, followed by phenology (Figure 6). Post-pollination and post-pollination barriers, namely pollen–stigma interactions, fruit set, and seed viability, had low or no impact on the RI (Figure 6). Moreover, some of the post-pollination barriers showed negative RI values, suggesting an advantage of interspecific crosses over within-species mating in a few species‘ pairs. Nonetheless, this can be attributed to the low sample size, whereas in some pairs, the number of available flowers for crosses was low (<10).

Total RI between pairs of species ranged between 0.078–0.99, with most of the species exhibiting high RI from each other (Table 2). RI values were relatively low (<0.5) between pairs of species from the northern region (*I. bismarckiana*, *I. haynei*, and *I. hermona*) and two of the southern species (*I. petrana* and *I. mariae*). 

### 2.6. Correlation between Total RI and Genetic Distance

We found relatively small genetic distances between species, as obtained from the phylogeny in Wilson et al. (2016; Figure 7). We did not find a significant correlation between phylogenetic distance and total RI for either RI calculations based on flowering time in TAUBG (Mantel test, Z = −0.435, *p* = 0.99) or for RI calculated based on flowering time in natural populations (Z= −0.437, *p* = 0.99). 

## 3. Discussion

Speciation as a dynamic and continuous process [8] has rarely been studied in a currently diverging species complex. Here, we estimated RI using pre- and post-zygotic reproductive barriers among eight *Oncocyclus Iris* species that form a young species complex in its early steps of speciation [33,37]. We found that pre-pollination barriers, mainly eco-geographic and phenological reproductive barriers, play a major role in driving speciation among these species. We also show that late-acting barriers have a negligible effect on reproductive isolation in this species complex. This study confirms the hypothesis that the role of pre-zygotic barriers, especially eco-geography, is more significant in speciation [3]. More specifically, we provide strong evidence for the hypotheses raised in previous studies that species divergence in the *Oncocyclus* species complex is driven by the eco-geographic barrier [35,36]. 

Long divergence time allows reproductive barriers to accumulate, resulting in an association between genetic divergence and the strength of RI. Studies in a few groups of species found a positive association of genetic distances with post-zygotic reproductive barriers [23,24]. In this study, we did not find such a correlation among the relatively young group of royal irises, thought to have evolved <2 million years ago [33]. Given that these are perennial plants, and that estimated generation time is 5–7 years due to strong seed dormancy and slow development [40]; Y. Sapir, unpublished, we speculate that the 400,000 generations or less were not sufficient to accumulate strong genetic divergence in association with the pre-zygotic reproductive barrier. For comparison, the annual California Jewelflower (*Streptanthus* spp.) is five million years old [25] and potentially accounts for five million generations, which is an order of magnitude longer than the royal irises. Indeed, California Jewelflowers showed a positive association between genetic distances and post-zygotic reproductive barriers [23,25]. The royal irises, evident by low sequence divergence, are a rapid-evolving group [29], which suggests that the first step of speciation, namely, eco-geography and phenology divergence, has only recently evolved.

The eco-geographic barrier was the strongest among the acting barriers we investigated. These results support our hypothesis and previous studies on the group [33,35,36]. Studies on other taxa revealed a similarly significant role of eco-geography as a reproductive barrier, e.g., [15,19,20,41,42]. The magnitude of eco-geography in speciation is important for several reasons. First, eco-geography is often the first acting barrier, thus leaving only a small potential contribution of later acting barriers to the total RI. Second, eco-geography can be associated with genetic incompatibilities among species due to local adaptation, leading to outbreeding depression after secondary contact. While this can drive reinforcement of RI through hybrid failure [43,44,45,46], due to the long generation time of the royal irises, we lack evidence for hybrid performance in the parental habitat. Nonetheless, it is evident that whether with or without genetic divergence, the eco-geographical barrier is the first to evolve during the speciation process in the royal irises, or, in some cases, it is flowering time. In royal irises, the presence of a single barrier (either eco-geography or phenology) that contributes to RI suggests that ecological divergence is the first step toward further divergence and speciation. 

We found that divergence in flowering time contributed to the RI of *I. lortetii* and *I. atropurpurea* to the other species in the complex. These results do not support our hypothesis that all the species in the complex would be isolated through the same mechanism. However, we hypothesize that the phenotypic isolation is a result of interaction with eco-geography, as it contributes to RI mainly among the northern species in the complex (Figure 1) and *I. atropurpurea*, suggesting that flowering time divergence contributes to RI where eco-geographic isolation is weaker. Previous studies on the group [35,36] did not investigate flowering time divergence as a possible barrier [35,36]. However, studies of other plant groups have found significant contributions of flowering time divergences to total RI [19,47,48]. Furthermore, studies in sympatric species showed that isolation by phenology might be driven by ecological divergence [49,50], suggesting that if an eco-geographic barrier becomes weaker phenology may emerge as the major driver of isolation.

We have found that post-pollination and post-zygotic barriers (pollen–pistil interactions, fruit set, and seeds viability) are practically absent among the *Oncocyclus* irises. This supports both our hypothesis and previous studies on the group [33,35,36]. Interestingly, our results are conceptually similar to a previous study that tested for within-species RI in *I. atropurpurea* [34]. There, the post-zygotic barriers (fruit- and seed sets) were associated mostly with ecological divergence. This implies that ecological adaptation is a primary driver of divergence and speciation also at the within-species level. This is in line with our hypothetical scenario of population divergence through local adaptation. This is in line with the findings of a previous study of RI among populations within species [34], which suggest that ecology drives speciation at multiple levels. 

We did not find a correlation between RI and the genetic divergence of the *Iris* species, contrary to our initial hypothesis and other studies on speciation. This suggests that the phylogenetic relationships among the species might be more complex than what was obtained by Wilson et al. (2016). We hypothesize that reproductive isolation might be reflected in small genomic regions or islands of speciation [51]. In addition, there is no clear morphological differentiation between these species, rather a clinal variation along the North-South aridity gradient, implying a lack of clear boundaries between species [32,35,36]. Although speciation studies are based on the biological species concept [2], other definitions of species boundaries, such as morphological or phylogenetic concepts, can be used [18]. Studying speciation requires the identification of species, but species delimitation in the *Oncocyclus* species complex depends on the species concept chosen [37]. The lack of distinct barriers and the failure of applying multiple species concepts to the *Oncocyclus* irises support the view of speciation as a continuum [8].

Our study of the incipient species complex in the course of speciation implies that species are not only found at different stages of isolation, but can also undergo multiple levels of divergence in several axes, such as pre-/post-zygotic, or intrinsic/extrinsic barriers [52]. We conclude that the *Oncocyclus* species of the genus *Iris* undergo rapid divergence along one evolutionary trajectory while exhibiting different stages of divergence in others. We propose that further studies, either in the royal irises species complex or in other species complexes, will facilitate our understanding of the speciation process.

## 4. Materials and Methods

### 4.1. Distribution and Environmental Data

The species of section *Oncocyclus* are distributed across the Middle East, including in Turkey, Iran, Syria, and Jordan, but due to limited information and lack of access to field sites, this study focused on the eight species growing in Israel and Palestine only. Distribution and flowering data were obtained from the Rotem (Israel Plants Information Center) database, maintained by Prof. A. Shmida, and from our own database (Y. Sapir, unpublished data). The data include coordinates of field observations, accurate to the resolution of 100 × 100 m. In addition, data include herbarium records dated as early as 1912. For most observations, data also included information on flowering phenology. 

Precipitation and temperature data were obtained from the 19 bioclimatic variables available in the WORLDCLIM database (http://www.worldclim.org/bioclim, accessed on 1 January 2017); [53]. The data were downloaded at the highest resolution (30 s~1 km^2^) and cropped to the range of the study area (34–29.4° N, 33.30–36.2° E) using the ‘Raster’ package for R [54]. GIS map layer of soil types was obtained from the geographical and ecological data center of the Israeli Nature and Park Authority and an elevation map was obtained from the GIS unit of The Hebrew University of Jerusalem. The resolution of the latter two layers was 50 × 50 m, much finer than the 1 × 1 km resolution of BIOCLIM data. Thus, the bioclimatic layers were interpolated using the Resample tool in the ArcGIS program to achieve the same resolution. 

### 4.2. Plant Material

For the flowering phenology, pollen–stigma interactions, and cross-experiments, we used an array of *Iris* plants of all Israeli species, maintained in the Tel-Aviv University Botanical Garden (TAUBG). The *Iris* collection consists of 720 plants of 8 species collected from natural populations across Israel between 1998 and 2017. The plants were grown in 10-liter polyethylene flexible containers filled with a mixture of 1:1 commercial potting soil and dune sand. The containers were placed on metal tables and organized by species. Each species was placed on at least two non-adjacent tables—accounting for possible spatial differences. Rhizomes of the plants were placed on top of the soil and covered with ~5 mm tuff stones. The plants were maintained in a net house to prevent exposure to pests and local pollinators and were annually treated with pesticides and fungicides. In addition to natural rain (583 mm of rain between December 2016 and April 2017), supplementary watering was applied twice a week between late October to late November 2016 using 2 l h^−1^ drippers for each plant. 

### 4.3. Eco-Geography

The eight studied *Iris* species are allopatric or parapatric (Figure 1); thus, we could not calculate a simple geographic overlap representing the realized niches. Instead, we calculated the overlap between potential niches. To calculate potential niche overlap between the species, we performed species distribution modeling (SDM), based on 2510 occurrence observations of all 8 species, and calculated the niche overlap between each species pair in geographic space. 

For the niche modeling, we used six bioclimatic variables from the WORLDCLIM data set: Bio1—annual mean temperature; Bio2—mean diurnal range; Bio4—temperature seasonality; Bio6—mean temperature of the coldest month; Bio12—annual precipitation; and Bio15—precipitation seasonality. These variables were chosen because in the studied region they had a lower correlation among them (<0.82; see Appendix A). In addition, we used GIS layers of elevation and soil type variables obtained from the Israeli Nature and Park Authority. 

To predict the potential niche of the irises, we used generalized linear models (GLMs) for the species distribution model [55]. This model is based on a linear regression between species presence and absences and the explanatory (environmental) variables [55]. The association between the predictors was linear, with no interactions. We generated absence points for each species using the ‘BIOMOD’ package for R [56] at a distance of 0.05 degrees from the perimeter of the distribution of each species, which is defined by the presence points [15,57]. The number of absence points for each species was determined as four times the number of presence points [57]. GLMs were constructed using the ‘SDM’ package for R [58]. While MaxEnt models are frequently used in species distribution modeling, they are better fitted to presence-only data, whereas GLM models are preferable when presence-absence data are available [59,60]. Because all populations of the irises in Israel are mapped in four decades of frequent surveys and are, thus, known for high accuracy, we assumed that, in high probability, there are no other iris populations in the absence area, i.e., the absence points are true-absence. With these types of data, GLM is the best-fitting model. To evaluate the accuracies of the GLMs, we used 70% of the data as training data and the remaining 30% as testing data. We then used the cross-validated area under the curve (AUC) and true skill statistics (TSS) estimators [58,61,62] to assess the accuracies of the SDM predictions (see Appendix A). In order to account for the possibility of pseudo-absence, we performed a MaxEnt model as well, which resulted in similar D values (but lower AUC values; see Appendix A).

To evaluate the overlap between the potential niches, we used the D index, which uses the probability of species X and species Y to occur in site *i* [38,63]: $$D\left(p_{X},p_{Y}\right)=1-\frac{1}{2}\underset{i}{\sum }\left|p_{X,i}-p_{Y,i}\right|$$

This index is widely used to evaluate niche overlap because of its simplicity and robustness [63]. For the calculation of D, we used the PhyloClim package [64].

### 4.4. Phenology

Phenological isolation between the species was measured by analyzing the overlap in flowering time. We measured the overlap of the probability of each species to flower each week along the flowering season, using the D index [65], similar to the calculation of eco-geographical overlap (see above). We monitored 530 flowers of all eight species, maintained in TAUBG (see above), and recorded for each plant the first day of flowering. Each recorded flower was marked to avoid multiple recordings. Monitoring flowering was conducted daily from early December 2016 to the end of April 2017. 

Flowering phenology data were also obtained from the ROTEM database of observations (see above). We recorded (as flowering) each record that was tagged, as start flowering, peak flowering, or end of flowering; a total of 746 observations were used. Flowering data were pooled across all years of observations (1979 to 2015) and outlier dates were manually removed. 

To compare flowering rates, we performed a time-to-event analysis by fitting the Cox proportional hazard model to the census data of flowering time for each species [66,67]. The day of flowering and the status of the flower (flowering/not flowering) were the explained variables and the species was used as an explanatory variable. The location in the net house (number of the table) and the year of observation in wild populations were added as random variables to the analyses of the net house data and observations in wild populations, respectively. For the analyses, we used the “Survival” package for R [68].

### 4.5. Pollen–Stigma Interactions

To estimate the RI induced by pollen–stigma interactions, we compared pollen germination of inter- and intra-specific crosses. Pollen recognition mechanisms are present on the stigma as well as along the style [69,70]. To obtain a sufficient sample size, we used *I. atropurpurea*, *I. petrana*, and *I. mariae* as pollen acceptors because of their higher abundance in the TAUBG collection. These species also represent different climate regions and different phylogenetic distances: *I. petrana* and *I. mariae* are desert species, while *I. atropurpurea* grows in the Mediterranean coastal habitat climate. In addition, *I. petrana* and *I. atropurpurea* are closer phylogenetically, while *I. mariae* is in a distant clade [29]. The experiment was conducted between February and April 2016, spanning the full flowering period of all three focal species.

Buds were covered by a mesh bag 2–3 days prior to flowering to prevent the natural pollination of bees that may have accidentally entered the net house. Prior to pollination treatment, stamens were removed. Pollen was collected from seven species available in the TAUBG collection: *I. atropurpurea*, *I. petrana*, *I. mariae*, *I. atrofusca*, *I. bismarckiana*, *I. hermona*, and *I. haynei*. An eighth species, *I. lortetii*, was also represented in the net house, but due to the later flowering period (see results), no between-species crosses were possible. We used a mix of pollen from different populations for each cross to control for within-species differentiation [34]. Each acceptor flower received three treatments, one for each of the three stigmas: one stigma received pollen of the same species (within species treatment), and the other two received pollen from two other species (between species treatments). Pollen was deposited on the stigma using a painting brush and the flowers were immediately recovered. Pollen was allowed to germinate for 24 h before the stigmas and the styles were removed from the flower using forceps and placed in a 50 mL tube. For *I. atropurpurea,* we performed 206 within-species pollinations and 525 between-species; for *I. mariae* we performed 162 within-species pollinations and 319 between-species pollinations; for *I. petrana* we performed 115 within-species pollinations and 247 between-species pollinations.

To stain and count pollen grains on stigmas, we used a modified protocol from Dafni et al. [71]. The collected stigmas were soaked in formaldehyde (42%): glacial acetic acid (35%): ethanol (96%) at a ratio of 5:5:90 respectively), and stored at 5 °C. Next, the stigmas were soaked in 50% ethanol for 24 h, rinsed thoroughly, and soaked in tap water for another 1–2 days. The stigmas were then moved to 4N NaOH solution for 24 h for tissue clearing and softening. The stigmas were rinsed and soaked again in tap water for one hour, placed on a microscope glass slide, stained with a few drops of 6% Aniline Blue (VWR Chemicals), and kept in a dark cool place for 4 h. The stigmas were examined under a fluorescent microscope (Olympus MVX10) with a UV filter under x 63 magnification. Three sections of each stigma were randomly selected for counting pollen grains and pollen tubes.

Crosses between *I. petrana* and *I. bismarckiana* were removed from the final analyses due to the small sample size (n < 5). To test for the effect of inter versus intraspecific origin of pollen on germination on the stigma, we used GLMs (linear response function) with each acceptor species and inter and intraspecific origins as the explanatory variables. The date of the cross was used as a covariate and the proportion of the germinated pollen out of the total number of pollen grains (arcsine transformed) was the explained variable. Model selection, to assess the importance of factors, was based on the Akaike information criterion (AIC).

### 4.6. Post-Zygotic Reproductive Barriers—Cross Experiment

To calculate the post-zygotic reproductive barrier, we compared fruit and seed sets between intra- and interspecific crosses. As in the pollen–stigma interactions experiment, we used *I. atropurpurea*, *I. petrana,* and *I. mariae* in the TAUBG collection as acceptor species due to their relatively high abundance in the collection. The experiment was performed between February and April 2013, spanning the full flowering period of these three acceptor species. Flowers of these species received pollen from all 8 Israeli *Iris* species (donors). Buds were covered with a mesh bag 2–3 days prior to anthesis, to prevent the natural pollination by bees that accidentally entered the net house. After anthesis, the flower was emasculated, and the anthers were used for reciprocal pollination. For each of the crosses, we used pollen from one population from a single species. The acceptor flower received one of two treatments: pollen of the same species (within species) or pollen of another species (between species). The pollination was performed by brushing the collected anthers on the three stigmas of the flower. The flowers were covered again until a fruit started to form, and the flower wilted. We performed 130 within-species and 60 between-species pollinations for *I. atropurpurea*; 43 within-species and 48 between-species pollinations for *I. mariae*; and 15 within-species and 27 between-species pollinations for *I. petrana*.

Fruits were collected after maturation, but before opening and dispersing seeds, 4–6 weeks after the cross. Fruits were cut with a utility knife and seeds were counted for each fruit. Inviable seeds were identified by their small size, relatively light color, and lack of endosperm (following Yardeni et al., 2016). For the analyses, we omitted crosses with *I. atrofusca*, *I. bismarckiana*, and *I. lortetii* as the donor species due to the small sample size (n < 5). We used GLM (linear response function) with acceptor species and inter- or intraspecific origins as the explanatory variables, and fruit set or the proportion of viable seeds out of total seeds (arcsine transformed) as the explained variables. Model selection was performed using AIC.

### 4.7. Reproductive Isolation

RI for each reproductive barrier was calculated using the approach proposed by Ramsey et al. [14] and Sobel and Chen [38]. Because the strength of reproductive isolation can be asymmetric [13,15], RI was separately calculated for each species with respect to each of the seven other species in the complex.

RI by eco-geographical separation was calculated using the index D, the probability of eco-geographical overlap of two species: $$RI_{ecogeography}=1-D=1-(1-\frac{1}{2}\underset{i}{\sum }\left|p_{x,i}-p_{y,i}\right|)$$

RI by phenology was similarly calculated using the index D, the probability of flowering time overlap of two species: $$RI_{Phenology}=1-D=1-\left(\frac{1}{2}\underset{i}{\sum }\left|p_{x,i}-p_{y,i}\right|\right)$$

RI by pollen–stigma interactions was calculated as the relative success of heterospecific pollen to germinate (following Sobel and Chen 2014 [38]): $$RI_{Pollen-stigma}=1-2\times \left(\frac{Gb}{Gw+Gb}\right)$$

*Gb* and *Gw* are the relative fractions of germinating pollen grains in between-species and within-species crosses, respectively. 

Post-zygotic RI was calculated for fruit-set and seeds resulting from the crosses: $$RI_{Fruit\;set}=1-2\times \left(\frac{Pb}{Pw+Pb}\right)\;\;\;RI_{seed\;set}=1-2\times \left(\frac{Pb}{Pw+Pb}\right)$$

*Pb* and *Pw* are the relative fractions of fruit produced from all crosses performed or the fraction of viable seeds out of the total amount of seeds produced in the crosses between-species and within-species, respectively. 

Total RI was calculated following Ramsey et al. (2003) [14]. First, we calculated the absolute contribution (AC) of the individual RI of each barrier relative to the previous one in the order in which it acts on isolation: $$AC_{n}=RI_{n}\left(1-\overset{n-1}{\underset{i=1}{\sum }}AC_{i}\right)$$

*AC_n_* is the absolute contribution of a barrier *n*, according to its order, *RI_n_* is the calculated RI of barrier *n*, and *AC_i_* is the absolute contribution of all the previous barriers. Next, we calculated the total RI (T) by summing all absolute contributions of all tested barriers: $$T=\overset{m}{\underset{i=1}{\sum }}AC_{i}$$

The total RI between all pairs of species was calculated twice, corresponding to the two parallel measures of the phenological barrier. For the first calculation, we used flowering data recorded in common-garden conditions in the net house. For the second calculation, we used flowering time data recorded in wild populations. Given that only three species could be used for post-pollination barriers analysis due to the small sample size, the calculations of total RI for a few pairs of species only partially represented the full sequence of barriers. For example, the total RI between *I. petrana* and *I. hermona* was calculated in full only for *I. petrana* as the pollen acceptor, while for the reverse, the total RI was calculated only for eco-geography and phenological barriers. 

### 4.8. Correlation between Total RI and Genetic Distance

The phylogenetic distance was estimated as the total branch length connecting each pair of species, i.e., the number of base changes separating each pair of species (Figure 2). These values were calculated based on the phylogeny tree obtained for the eight species using OneTwoTree web server [39], which retrieved sequences reported in Wilson, Padiernos, and Sapir [29]. We used the Mantel test (package “Vegan” for R; Oksanen et al., 2017) to test for correlation between paired phylogenetic distance and total RI.

## 5. Conclusions

Studying speciation in a species complex provides insight into the process of incipient speciation. Our study of the royal irises species complex contributes to the understanding of the process/the results support the hypothesis that eco-geographical divergence, and to some extent also flowering time, govern the first step of speciation in the royal irises. While the pattern of the eco-geographical reproductive barrier is similar across most species pairs in the complex, a few also demonstrated divergence in flowering time. Interestingly, this phenological isolation barrier was stronger for sympatric or parapatric species. These results shed light on how species boundaries are developed and maintained mainly through early-acting barriers and provide that studying speciation in a species complex could uncover the interplay between different barriers to maintain species coherence.

## Figures and Tables

**Figure 1 plants-11-03306-f001:**
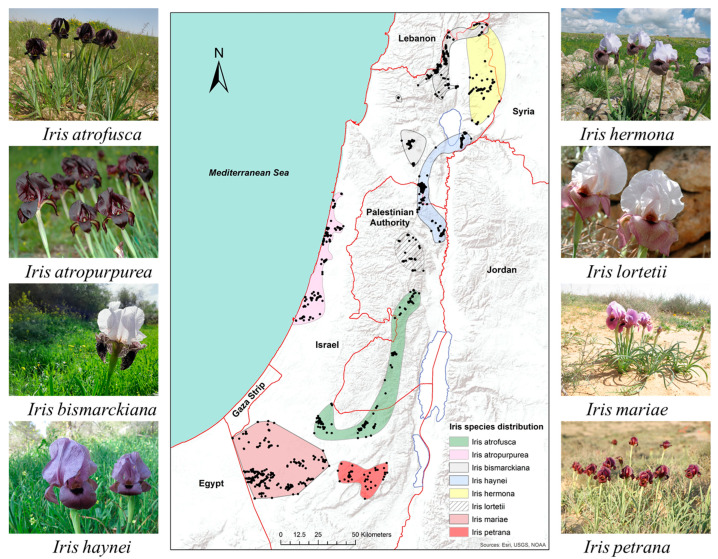
Distribution of the eight species of royal irises in Israel and Palestine. Points are all confirmed observations, starting in the early 1900s, including populations extinct due to human disturbances.

**Figure 2 plants-11-03306-f002:**
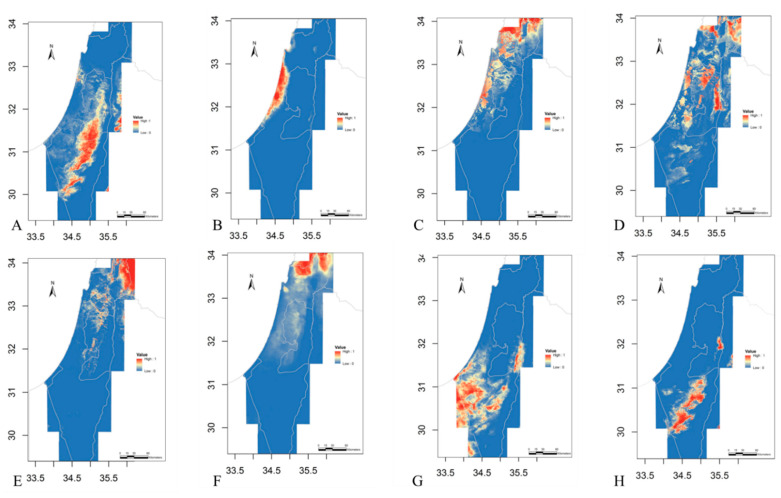
The potential distribution of the eight Israeli *Oncocyclus* irises, as predicted by the SDMs. The probability to find a species in a certain environment range from 0 (dark blue), through 0.5 (yellow) to 1 (red). (**A**) *Iris atrofusca*; (**B**) *I. atropurpurea*; (**C**) *I. bismarckiana*; (**D**) *I. haynei*; (**E**) *I. hermona*; (**F**) *I. lortetii*; (**G**) *I. mariae*; (**H**) *I. petrana*.

**Figure 3 plants-11-03306-f003:**
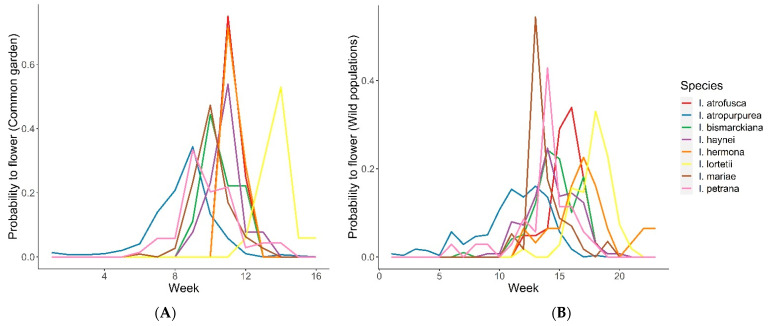
Flowering time, expressed as the probability of flowering in (**A**) common garden conditions in TAUBG, and (**B**) observations from wild populations.

**Figure 4 plants-11-03306-f004:**
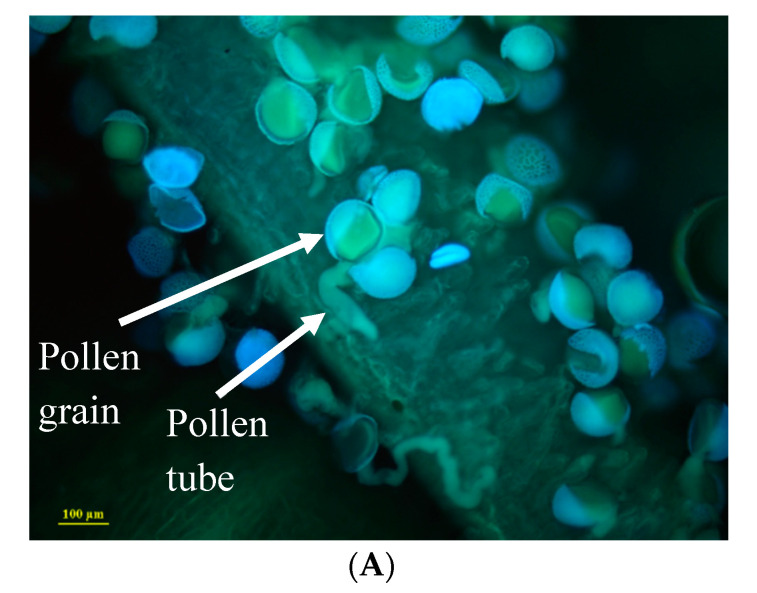
(**A**) Pollen grains and tubes of *Iris atropurpurea* on the stigma of a conspecific plant, stained with aniline blue. (**B**) The proportion of germinated pollen from within (yellow) and between (purple) the species origins on the stigmas of three recipient species. The differences between treatments were not significant (*p* > 0.05) for all three recipient species.

**Figure 5 plants-11-03306-f005:**
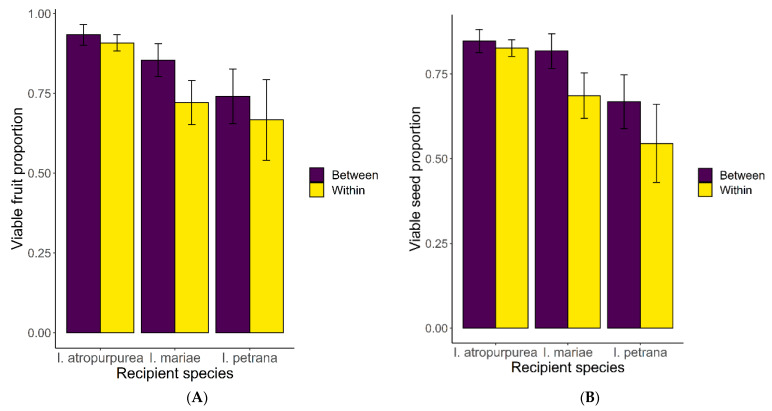
The proportion of viable fruits (**A**) and seeds (**B**) within (red) and between (grey) species crosses, in three recipient species. The differences between treatments were not significant (*p* > 0.05) for all three recipient species.

**Figure 6 plants-11-03306-f006:**
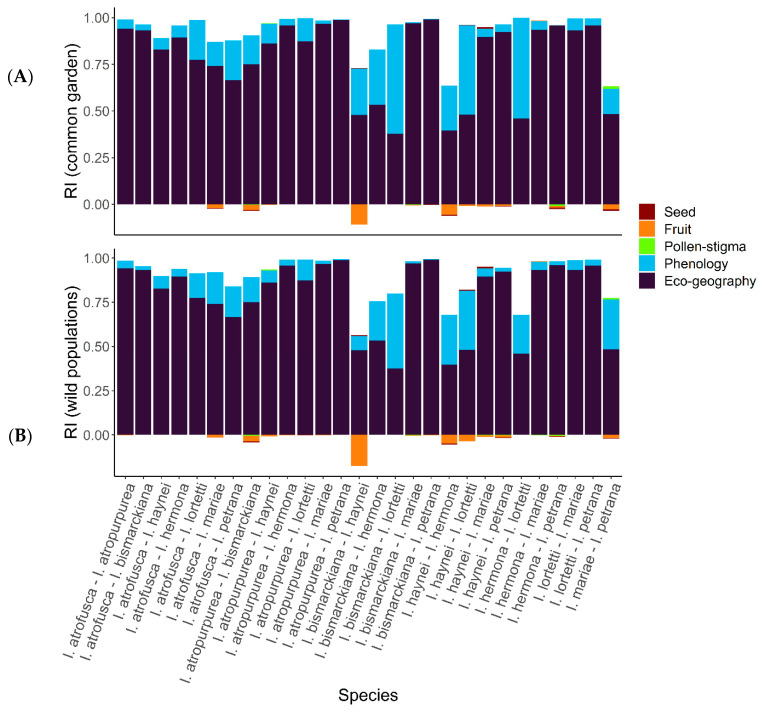
The relative contribution of individual reproductive barriers to total reproductive isolation (RI) between pairs of species. (**A**) RI calculated with phenology measured in a controlled environment at the net house in the Tel Aviv University Botanic Garden. (**B**) RI calculated with phenology recorded in wild populations.

**Figure 7 plants-11-03306-f007:**
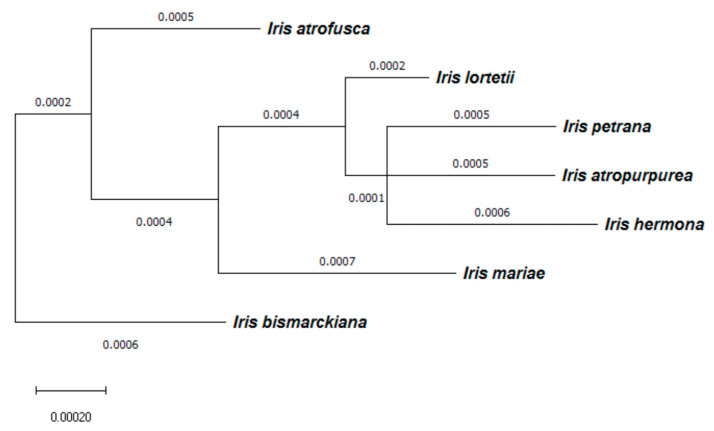
Phylogenetic tree of the 8 Israeli *Iris* species. Phylogeny based on data retrieved from the OneTwoTree web server [39]. Scale: number of SNPs in 1000 sequence bases.

**Table 1 plants-11-03306-t001:** D values between the eight *Iris* species. The values are calculated for the species in the rows, with respect to the species in the columns. Shades denote the extent of the eco-geographical overlap, i.e., black for high overlap and white for complete isolation.

	*I. atrofusca*	*I. atropurpurea*	*I. bismarckiana*	*I. haynei*	*I. hermona*	*I. lortetii*	*I. mariae*
*I. atropurpurea*	0.02	1.00					
*I. bismarckiana*	0.06	0.06	1.00				
*I. haynei*	0.15	0.03	0.36	1.00			
*I. hermona*	0.16	0.04	0.47	0.46	1.00		
*I. lortetii*	0.20	0.04	0.66	0.37	0.45	1.00	
*I. mariae*	0.12	0.01	0.02	0.03	0.05	0.04	1.00
*I. petrana*	0.22	0.00	0.01	0.04	0.05	0.06	0.34

**Table 2 plants-11-03306-t002:** Reproductive isolation (RI) values among eight Israeli *Iris* species. Above the diagonal—RI values calculated with phenology data from net-house (common garden); Below the diagonal—RI values calculated with phenology data from wild populations (see METHODS).

	*I. atrofusca*	*I. atropurpurea*	*I. bismarckiana*	*I. haynei*	*I. hermona*	*I. lortetii*	*I. mariae*	*I. petrana*
*I. atrofusca*		0.99	0.96	0.89	0.96	0.99	0.85	0.88
*I. atropurpurea*	0.99		0.87	0.97	0.99	1.00	0.99	0.99
*I. bismarckiana*	0.95	0.85		0.62	0.83	0.96	0.97	0.99
*I. haynei*	0.90	0.93	0.39		0.57	0.95	0.94	0.95
*I. hermona*	0.94	0.99	0.75	0.63		1.00	0.98	0.94
*I. lortetii*	0.91	0.99	0.80	0.78	0.68		1.00	1.00
*I. mariae*	0.91	0.99	0.98	0.94	0.98	0.99		0.60
*I. petrana*	0.84	0.99	0.99	0.93	0.97	0.99	0.75	

## Data Availability

The data presented in this study are openly available in Dryad at doi.org/10.5061/dryad.ksn02v77v.

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
