# Peer review of "Eco-Geography and Phenology Are the Major Drivers of Reproductive Isolation in the Royal Irises, a Species Complex in the Course of Speciation"

_plants, 2022, doi:10.3390/plants11233306_

Round 1

Reviewer 1 Report

All comments are in the attached file

Author Response

Responses to all comments are posted in the file attached. 

Reviewer 2 Report

Dear authors,

This manuscript is suitable for this journal.

Line 47-48: RI abbreviation should after the full name for the first time usage.

Please edit the format of words into Italic for all species name have been mentioned in this manuscript.

Please provide better resolution for Figure 7 for A and B graphs species name.

After discussion, will you want to write a small paragraph for conclusion? (independent from the discussion)

Finally, do you mind to write more clearly for your replicates for each experiment (methods)? For example, how many replicates for each performance of Pollen-stigma interactions and Reproductive isolation?

Author Response

This manuscript is suitable for this journal.

Response: We thank the reviewer for this encouraging and positive opinion.

Line 47-48: RI abbreviation should after the full name for the first time usage.

Response: RI abbreviation appear in full in its first time usage (line 33).

Please edit the format of words into Italic for all species name have been mentioned in this manuscript.

Response: All species names were italicized throughout the manuscript.

Please provide better resolution for Figure 7 for A and B graphs species name.

Response: We improved the graphics of all graphs in the manuscript.

After discussion, will you want to write a small paragraph for conclusion? (independent from the discussion)

Response: Thank you for the idea. We have added a short summary and conclusion paragraph. 

Finally, do you mind to write more clearly for your replicates for each experiment (methods)? For example, how many replicates for each performance of Pollen-stigma interactions and Reproductive isolation?

Response: Sample sized were added.

Round 2

Reviewer 1 Report

All suggestions have been answered.

Excellent ideas added, they have complemented the article.